# Effect of an editorial intervention to improve the completeness of reporting of randomised trials: a randomised controlled trial

David Blanco [1,2] Sara Schroter [3] Adrian Aldcroft [3] David Moher [4]
Isabelle Boutron,[1] Jamie J Kirkham,[5] Erik Cobo[1]

JJK and EC are joint senior authors.

¹Statistics and Operations Research Department, Universitat Politecnica de Catalunya, Barcelona, Spain
²CRESS, INSERM, INRA, Université de Paris, Paris, France
³The BMJ, London, UK
⁴Centre for Journalology, Clinical Epidemiology Program, Ottawa Hospital Research Institute, Ottawa, Ontario, Canada
⁵Centre for Biostatistics, Manchester Academic Health Science Centre, Manchester University, Manchester, UK

**Correspondence to**
David Blanco;
david.blanco.tena@upc.edu

## ABSTRACT

**Objective** To evaluate the impact of an editorial intervention to improve completeness of reporting of reports of randomised trials.
**Design** Randomised controlled trial (RCT).
**Setting** *BMJ Open*'s quality improvement programme.
**Participants** 24 manuscripts describing RCTs.
**Interventions** We used an R Shiny application to randomise manuscripts (1:1 allocation ratio, blocks of 4) to the intervention (n=12) or control (n=12) group. The intervention was performed by a researcher with expertise in the content of the Consolidated Standards of Reporting Trials (CONSORT) and consisted of an evaluation of completeness of reporting of eight core CONSORT items using the submitted checklist to locate information, and the production of a report containing specific requests for authors based on the reporting issues found, provided alongside the peer review reports. The control group underwent the usual peer review.
**Outcomes** The primary outcome is the number of adequately reported items (0–8 scale) in the revised manuscript after the first round of peer review. The main analysis was intention-to-treat (n=24), and we imputed the scores of lost to follow-up manuscripts (rejected after peer review and not resubmitted). The secondary outcome is the proportion of manuscripts where each item was adequately reported. Two blinded reviewers assessed the outcomes independently and in duplicate and solved disagreements by consensus. We also recorded the amount of time to perform the intervention.
**Results** Manuscripts in the intervention group (mean: 7.01; SD: 1.47) were more completely reported than those in the control group (mean: 5.68; SD: 1.43) (mean difference 1.43, 95% CI 0.31 to 2.58). We observed the main differences in items 6a (outcomes), 9 (allocation concealment mechanism), 11a (blinding) and 17a (outcomes and estimation). The mean time to perform the intervention was 87 (SD 42) min.
**Conclusions** We demonstrated the benefit of involving a reporting guideline expert in the editorial process. Improving the completeness of RCTs is essential to enhance their usability.
**Trial registration number** NCT03751878.

### Strengths and limitations of this study

► We used a randomised controlled trial design and implemented the intervention in a real editorial context.
► Outcome assessment was blinded and in duplicate.
► We focused only on eight items of one reporting guideline (Consolidated Standards of Reporting Trials).
► The intervention was performed in only one journal.

## INTRODUCTION

The lack of transparency and accuracy of research reports has been pointed out as one of the main factors causing research waste.[1] Adequate reporting allows researchers to replicate results, generate new hypothesis or compare the results of different studies; allows healthcare professionals to make clinical decisions; allows governments to change public policies; and helps patients to be aware of what healthcare options they have.[2]

Reporting guidelines (RGs) are sets of minimum recommendations for authors, usually in the form of a checklist, on how to report research methods and findings so that no relevant information is omitted.[2] Since the inception in 1996 of the Consolidated Standards of Reporting Trials (CONSORT) for the reporting of randomised controlled trials (RCTs),[3] hundreds of RGs for different study types, data, and preclinical and clinical areas have been developed.[4] CONSORT is currently one of the most well-established RGs and has been revised and updated twice.[5 6]

Most RGs have not been evaluated as to whether they actually improve completeness of reporting. Even for those that have been shown to be beneficial, such as CONSORT, the degree of author adherence is poor.[7] For this reason, a range of interventions aimed to improve adherence to RGs have been

proposed, and the impact of some of these on completeness of reporting has been evaluated. A recent scoping review identified and classified 31 interventions targeting different stakeholders, including authors, peer reviewers, journal editors, medical schools and ethics boards.[8] Among these, only four were assessed in RCTs and their effects were varied.[9–12] Most of the studies included in the scoping review described observational studies that evaluated the pooled effect of different journal strategies, which ranged from making available editorial statements that endorse certain RGs, recommending or requiring authors to follow RGs in the 'Instructions to authors', and requiring authors to submit a completed RG checklist together with the manuscript. However, these actions have been shown not to have the desired effect.[13–16] In contrast, completeness of reporting improved remarkably when editors were in the process of checking adherence to RGs.[17]

Recently, many biomedical journals have opted for requiring the submission of RG checklists alongside the manuscript. While sometimes checking these is delegated to peer reviewers, journal editors generally report that this task goes beyond the role of these and that it may even decrease the quality of peer review reports.[18] If checking reporting issues becomes a standard exercise for peer reviewers, some editors are afraid that peer reviewers may be less likely to comment on important aspects of a manuscript, such as its importance, novelty and relevance. Involving trained experts or administrative staff could be a way to make the most of this editorial strategy.[18]

### Study objectives

We describe an RCT to evaluate the effect of an editorial intervention performed by a researcher with expertise in CONSORT on the completeness of reporting of trials submitted to *BMJ Open*, compared with the standard peer review process.

## METHODS

### Trial design and study setting

This was a two-arm, parallel, randomised trial (1:1 allocation ratio) conducted in collaboration with *BMJ Open*, an open-access general medical journal (published by the BMJ Publishing Group) that requests the submission of completed CONSORT checklists for RCTs. Prior to recruitment, we registered the study in ClinicalTrials.gov and uploaded the study protocol.[19]

### Eligibility criteria

Manuscripts were eligible for inclusion if (1) they were original research articles reporting the results of an RCT submitted to *BMJ Open*, (2) they had passed the first editorial filter and had been subsequently sent out for peer review, and (3) the authors of these manuscripts had provided a completed CONSORT checklist as part of the submission process. Apart from the standard,

---

**Box 1  Example of report reflecting the reporting issues found**

Please make the following revisions:
► For CONSORT item 8a ('Method used to generate the random allocation sequence'), please report the exact method you used to generate the random allocation sequence.
  – Example from CONSORT: 'Randomization sequence was created using Stata M.N (StataCorp, College Station, TX) statistical software'.
► For CONSORT item 11a ('If done, who was blinded after assignment to interventions and how'), please specify in 'Trial design and setting' who was blinded in the study and do not just state that it was a double-blind randomised trial.
  – Example from CONSORT: 'Whereas patients and physicians allocated to the intervention group were aware of the allocated arm, outcome assessors and data analysts were kept blinded to the allocation'.

CONSORT, Consolidated Standards of Reporting Trials.

---

two-arm, parallel RCTs, which are covered by the standard CONSORT guidelines,[20] we also included RCTs that require the use of the official CONSORT extensions for different design aspects (cluster,[21] non-inferiority and equivalence,[22] pragmatic,[23] N-of-1 trials,[24] pilot and feasibility,[25] and within-person trials)[26] and intervention types (herbal,[27] non-pharmacological,[28] acupuncture[29] and Chinese herbal medicine formulas[30]) in all areas of clinical research. We excluded studies that claimed to be RCTs but used deterministic allocation methods and secondary trial analysis studies.

### Interventions

We designed a three-step intervention based on the results of our previous work,[8 18] ensuring no disruption to usual editorial procedures. The lead investigator (DB), a PhD student with a background in statistics who had worked for 2 years on the topic of improving adherence to RGs and who had expertise in the content of CONSORT, performed the intervention. First, he assessed completeness of reporting of eight core CONSORT items (see the following paragraph) using the submitted checklist to locate the information corresponding to each item. Second, he produced a standardised report containing precise requests to be addressed by authors. This report included a point by point description of the reporting issues found, requests to the authors to include the missing information (see example in box 1), as well as examples extracted from the CONSORT Explanation and Elaboration (E&E) document.[20] Finally, DB uploaded the report to the manuscript tracking system of the journal (ScholarOne) to make it accessible to the manuscript handling editor, who included this additional report in the decision letter to authors alongside the standard peer review reports. Manuscripts randomised to the control group underwent the usual peer review process. In figure 1, we display a schema of the study design.

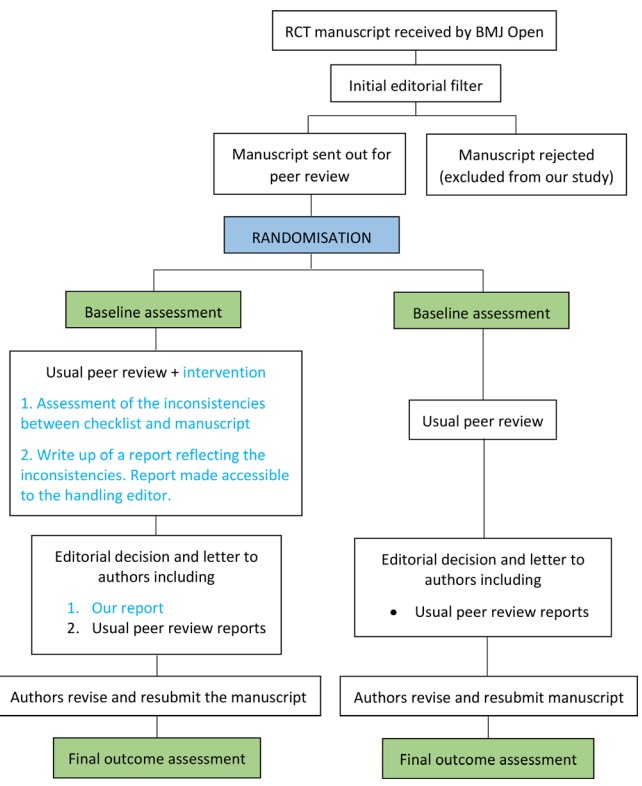

**Figure 1** Schema of the study design. RCT, randomised controlled trial.

The intervention was focused on eight core CONSORT items (see box 2) which are essential for researchers evaluating the risk of bias of RCTs when conducting systematic reviews[31] and which are usually poorly reported.[32]

We considered an item as adequately reported if all subparts of it were adequately reported, according to the

---

**Box 2   Core CONSORT items considered**

Five items in the methods section:
- ► Item 6a ('Completely defined pre-specified primary and secondary outcome measures, including how and when they were assessed').
- ► Item 8a ('Method used to generate the random allocation sequence').
- ► Item 9 ('Mechanism used to implement the random allocation sequence (such as sequentially numbered containers), describing any steps taken to conceal the sequence until interventions were assigned').
- ► Item 11a ('If done, who was blinded after assignment to interventions (for example, participants, care providers, those assessing outcomes) and how').
- ► Item 11b ('If relevant, description of the similarity of interventions').

Three items in the results section:
- ► Item 13a ('For each group, the numbers of participants who were randomly assigned, received intended treatment, and were analysed for the primary outcome').
- ► Item 13b ('For each group, losses and exclusions after randomisation, together with reasons').
- ► Item 17a ('For each primary and secondary outcome, results for each group, and the estimated effect size and its precision (such as 95% confidence interval)').

CONSORT, Consolidated Standards of Reporting Trials.

---

CONSORT E&E document[20] and the corresponding E&E documents for the extensions considered. For example, for CONSORT item 6a ('Completely defined pre-specified primary and secondary outcome measures, including how and when they were assessed'), we required the following subparts to be adequately reported: (1) identified and completely defined primary and secondary outcomes, (2) analysis metric and methods of aggregation for each outcome, and (3) time points for each outcome.

The items corresponding to CONSORT extensions were assessed in addition to the standard CONSORT items. For example, we expected authors of a cluster randomised trial evaluating a pharmacological treatment to be using the standard CONSORT checklist for all eight items and the cluster extension for items 6a, 9, 13a, 13b and 17a. In contrast, the items requested by the pilot and feasibility extension substituted the standard CONSORT items, as specified in its E&E document.[25] Once the recruitment had begun, we decided to discard the extension for non-pharmacological interventions as it was not being requested by the editors nor sent by the authors.

In online supplementary file 1, we present further details on the rules we used to deal with not applicable items and with certain aspects of specific items.

### Outcomes

- ► Primary outcome: the mean score for completeness of reporting, defined as the mean number of adequately reported items in the first revised manuscript (0–8 scale).
- ► Secondary outcome: proportion of manuscripts where each item was adequately reported.

In the design phase of the study, we considered two potential scenarios where included manuscripts could potentially be lost to follow-up: (1) when editors rejected a manuscript after peer review and (2) when authors did not return the revised manuscript within the period requested by the handling editor after a 'Minor revision' or 'Major revision' editorial decision (14 and 28 days, respectively, plus, if necessary, the extra time that the editor considered appropriate). In the 'Statistical methods' section, we report the methods used to impute the study outcomes for lost to follow-up articles.

Outcome evaluation was performed independently and in duplicate by two senior researchers (EC, JJK) who were blinded to manuscript allocation and had experience as authors and reviewers of RCTs. They also assessed outcomes at baseline. In cases where a manuscript was rejected after the first round of peer review, assessors could only evaluate it at baseline. However, they were not aware of the fate of that manuscript until after they had completed that evaluation. More details about the outcome assessment process can be found in online supplementary file 2.

For each of the manuscripts in the intervention group, we also recorded the amount of time it took the lead investigator to perform the intervention.

## Harms

We analysed whether our intervention caused the following unintended effects: higher proportion of manuscript rejections after the first round of peer review and delays in the submission of the revised manuscripts by authors.

## Pilot work

To inform the sample size calculation, the lead investigator assessed 12 randomly selected RCTs published in *BMJ Open* between April 2018 and September 2018. The proportions of adequately reported items observed in these manuscripts were used to estimate the scores for completeness of reporting of the manuscripts in the control group (usual peer review).

Furthermore, outcome assessors (EC, JJK) practised the evaluation of completeness of reporting by assessing 6 of the 12 RCTs that were mentioned.

## Power analysis

According to the assessment described in the 'Pilot work' section, the estimated probabilities that manuscripts in the control group adequately reported 0, 1, 2,…, and 8 items were 0, 0, 0, 0, 0, 0.17, 0.33, 0.33 and 0.17, respectively. With the intervention, we aimed to bring this distribution to 0, 0, 0, 0, 0, 0, 0, 0.5 and 0.5. In other words, manuscripts in the intervention group were expected to be adequately reporting seven or eight items 50% of the time, respectively.

In order to relax the strong required assumptions behind using a t-test for a reduced sample size, we used bootstrapping, a simple yet powerful non-parametric technique.[33] First, given the probability distributions mentioned, we performed 10 000 simulations of the scores of n manuscripts. We resampled each of these simulations 10 000 times in order to calculate the 95% CI of the mean difference between groups. Finally, we calculated the study power by counting for how many of the 10 000 simulations the lower limit of this 95% CI was over 0.

Choosing a sample size of 24 manuscripts (12 per arm) and following the procedure above gave us 90% power (alpha=0.05, two-tailed). The R code used can be found in online supplementary file 3, script A.

## Randomisation and blinding

Prior to recruitment of manuscripts, DB screened automated reports listing original research submissions to *BMJ Open* on ScholarOne, daily, including their identification (ID), date of submission, title, abstract and different parameters related to their peer review status. RCTs were identified for possible inclusion based on the title and abstract and then checked against our eligibility criteria until the desired sample size was achieved.

Every time a manuscript met our eligibility criteria, DB introduced its ID into an R Shiny application[34] created by a senior statistician (JAG) (see online supplementary file 3, script B), which randomised the manuscript to the intervention or the control group (1:1 allocation ratio,

blocks of 4). Manuscripts were stratified according to whether there was an applicable CONSORT extension for that study or not. To avoid allocation bias, each ID could only be introduced once.

As part of the usual submission process, all authors are informed that the BMJ Publishing Group has a quality improvement programme and their manuscript might be entered into a study. However, authors of included manuscripts were not explicitly informed that their manuscripts were part of an RCT.

Outcome assessors were blinded to allocation and to each other's evaluation. Handling editors of the included manuscripts and the investigator performing the intervention (DB) were not blinded.

## Statistical methods

We carried out statistical analysis using R V.3.6.0.[35]

For the primary outcome, we adjusted a linear regression model with the baseline score of the manuscript as the only covariate. We calculated the 95% CI using bootstrapping (see online supplementary file 3, script C).

The main analysis of the primary outcome was intention-to-treat: all manuscripts were included in this analysis regardless of whether they were lost to follow-up. We imputed the scores of lost to follow-up manuscripts with a value of $8-b$, where $b$ was the baseline score of the manuscript. This imputation strategy aimed to reflect the fact that rejecting RCTs of low baseline quality could be considered an editorial success. In addition, we assessed the sensitivity of the results by carrying out a complete case analysis and analysing the best case (manuscripts in the intervention group reached the maximum score and controls did not improve) and worst case (manuscripts in the intervention group did not improve and controls reached the maximum score) scenarios.

We did not plan any subgroup analysis (see protocol[19]) and so none is reported.

## Deviations from the protocol

The last criterion (3: authors of the manuscripts had provided a completed CONSORT checklist) was not included in the first version of the protocol, but we implemented it before recruitment started. The reason was that, despite the submission of the CONSORT checklist for trials being mandatory, we observed that handling editors were occasionally overlooking this requirement and sending out manuscripts of trials for peer review that did not include one. Second, we initially used a t-test to calculate the study power and planned to use it for the primary outcome analysis. However, for the reasons described in the 'Power analysis' section, we used a bootstrap approach and the study power increased from the 85% stated in the protocol to 90%. Third, we decided to assess the baseline scores for completeness of reporting for the included manuscripts in order to adjust for these in the primary outcome analysis. With this we tried to avoid that a difference in the baseline scores between the two groups could make the intervention seem to have a

larger or smaller effect than it actually had. Finally, we added a best-case and worst-case scenario analysis to assess the sensitivity of the primary outcome results.

## Reporting guidelines

We report this manuscript in accordance with CONSORT 2010.[6]

## Patient and public involvement

Patients were not study participants and were not involved in setting the research question, designing the study, in the conduct of the study or in the interpretation of the results.

## RESULTS

Between 31 October 2018 and 4 April 2019, we screened 62 manuscripts that described RCTs submitted to *BMJ Open*. Among these, we excluded 38 either because they were rejected without peer review (n=34) or because the authors did not provide the CONSORT checklist (n=4). We randomised the remaining 24 to the intervention (n=12) or control (n=12) groups. Six (25%) manuscripts were lost to follow-up (intervention n=3, control n=3) as they were rejected after the first round of peer review and therefore not returned to authors for revision (scenario 1 in the Outcomes section). No manuscripts were lost to follow-up in scenario 2 as all authors returned the revised manuscripts within the given time. Therefore, 18 manuscripts (intervention n=9, control n=9) were revised by authors. Figure 2 shows the flow diagram of the study.

Most manuscripts (n=19, 79%) required at least one extension: non-pharmacological (intervention n=10, control n=8), pilot and feasibility (n=3, n=4), and cluster

| Table 1 | Baseline characteristics of the included randomised controlled trials | |
|---|---|---|
| | **Intervention (n=12)** | **Control (n=12)** |
| Study design | | |
| Standard parallel-group | 7 (58%) | 7 (58%) |
| Cluster | 2 (17%) | 1 (8%) |
| Pilot and feasibility | 3 (25%) | 4 (33%) |
| Type of intervention | | |
| Pharmacological | 2 (17%) | 4 (33%) |
| Non-pharmacological | 10 (83%) | 8 (67%) |
| Behavioural | 4 (33%) | 3 (25%) |
| E-health and tele-health strategies | 3 (25%) | 2 (17%) |
| Medical devices | 2 (17%) | 1 (8%) |
| Surgery | 0 (0%) | 1 (8%) |
| Others | 1 (8%) | 1 (8%) |
| Single-centre or multicentre | | |
| Single-centre | 8 (67%) | 5 (42%) |
| Multicentre | 4 (33%) | 7 (58%) |
| Number of participants | | |
| ≤50 | 5 (42%) | 2 (17%) |
| >50 and ≤100 | 3 (25%) | 7 (58%) |
| >100 | 4 (33%) | 3 (25%) |
| Registered in a trial registry | | |
| Yes | 11 (92%) | 11 (92%) |
| No | 1 (8%) | 1 (8%) |
| First author's affiliation | | |
| Asia | 3 (25%) | 3 (25%) |
| UK | 3 (25%) | 5 (42%) |
| Europe | 2 (17%) | 3 (25%) |
| USA | 2 (17%) | 0 (0%) |
| Australia | 2 (17%) | 0 (0%) |
| Brazil | 0 (0%) | 1 (8%) |
| Sponsorship | | |
| Investigator-initiated | 12 (100%) | 10 (83%) |
| Industry-initiated | 0 (0%) | 2 (17%) |

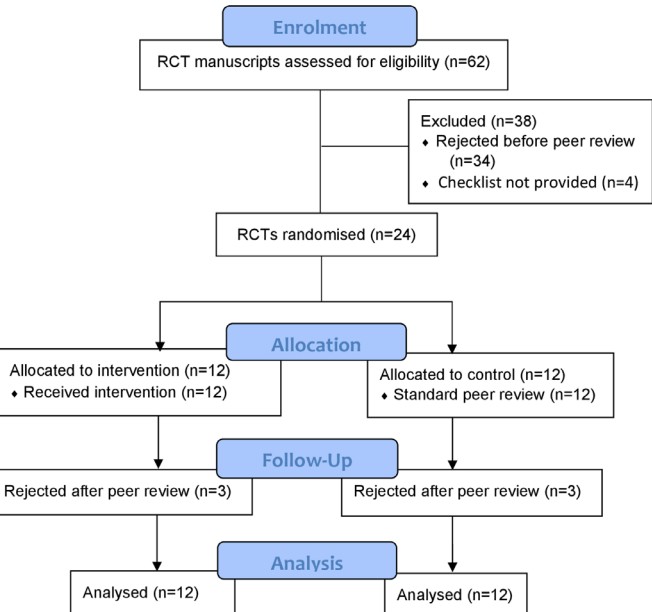

**Figure 2** CONSORT flow diagram. CONSORT, Consolidated Standards of Reporting Trials; RCT, randomised controlled trial.

(n=2, n=1). Table 1 displays the baseline characteristics of the included manuscripts.

The mean (SD) baseline score for completeness of reporting (0–8 scale) prior to peer review in the intervention (n=12) and control (n=12) groups was 4.35 (1.88) and 4.85 (1.79), respectively. The mean (SD) baseline score of the manuscripts that later passed the first round of peer review (n=18) was much more complete (scores almost double) than those that were rejected after the first round of peer review (n=6): 5.23 (1.35) vs 2.68 (1.75).

**Table 2** Scores for completeness of reporting scores in the control and intervention groups

| Outcome | Intervention group Mean (SD) | | Control group Mean (SD) | | Mean difference in final scores* (95% CI) |
|---|---|---|---|---|---|
| | Baseline | Final | Baseline | Final | |
| Completeness of reporting (0–8 scale) with imputation (n=24) | 4.35 (1.88) | 7.01 (1.47) | 4.85 (1.79) | 5.68 (1.43) | 1.43 (0.31 to 2.58) |
| Completeness of reporting (0–8 scale) without imputation (complete case analysis, n=18) | 5.01 (1.32) | 7.45 (1.00) | 5.46 (1.41) | 5.90 (1.35) | 1.75 (0.80 to 2.75) |
| Completeness of reporting (0–8 scale) in the best-case scenario (n=24) | 4.35 (1.88) | 7.59 (0.89) | 4.85 (1.79) | 5.18 (1.89) | 2.62 (1.49 to 3.65) |
| Completeness of reporting (0–8 scale) in the worst-case scenario (n=24) | | 6.18 (2.61) | | 6.43 (1.49) | 0.03 (−1.45 to 1.63) |

*Adjusted for baseline score.

### Primary outcome

For the intention-to-treat analysis (n=24), the manuscripts that received the intervention were more completely reported than the ones that underwent the standard review process (intervention group: mean 7.01 (SD 1.47) vs control group: mean 5.68 (SD 1.43)). After adjusting for the baseline score, the mean difference in scores between the two groups was 1.43 (95% CI 0.31 to 2.58); the manuscripts in the intervention group reported on average 1.43 (out of 8) items more adequately than those receiving the standard peer review. Regarding the sensitivity analysis, for the complete case (n=18) the mean (SD) scores for the intervention and control groups were 7.45 (1.00) and 5.90 (1.35), giving an adjusted difference of 1.75 (95% CI 0.80 to 2.75). The best-case and worst-case scenario analysis (n=24) led to adjusted differences of 2.62 (95% CI 1.49 to 3.65) and 0.03 (95% CI −1.45 to 1.63), respectively. Table 2 summarises these results.

Figure 3 shows the evolution of the 18 manuscripts that were revised and resubmitted. From the nine manuscripts in the intervention group, six of them achieved the maximum score and another two improved. In contrast, the only manuscript in the control group that reached the maximum score already had that score at baseline. Three manuscripts in the control group slightly improved (1, 1 and 2 points, respectively). We identified that three out of four of these improvements were the result of comments made by the standard peer reviewers, rather than the authors themselves.

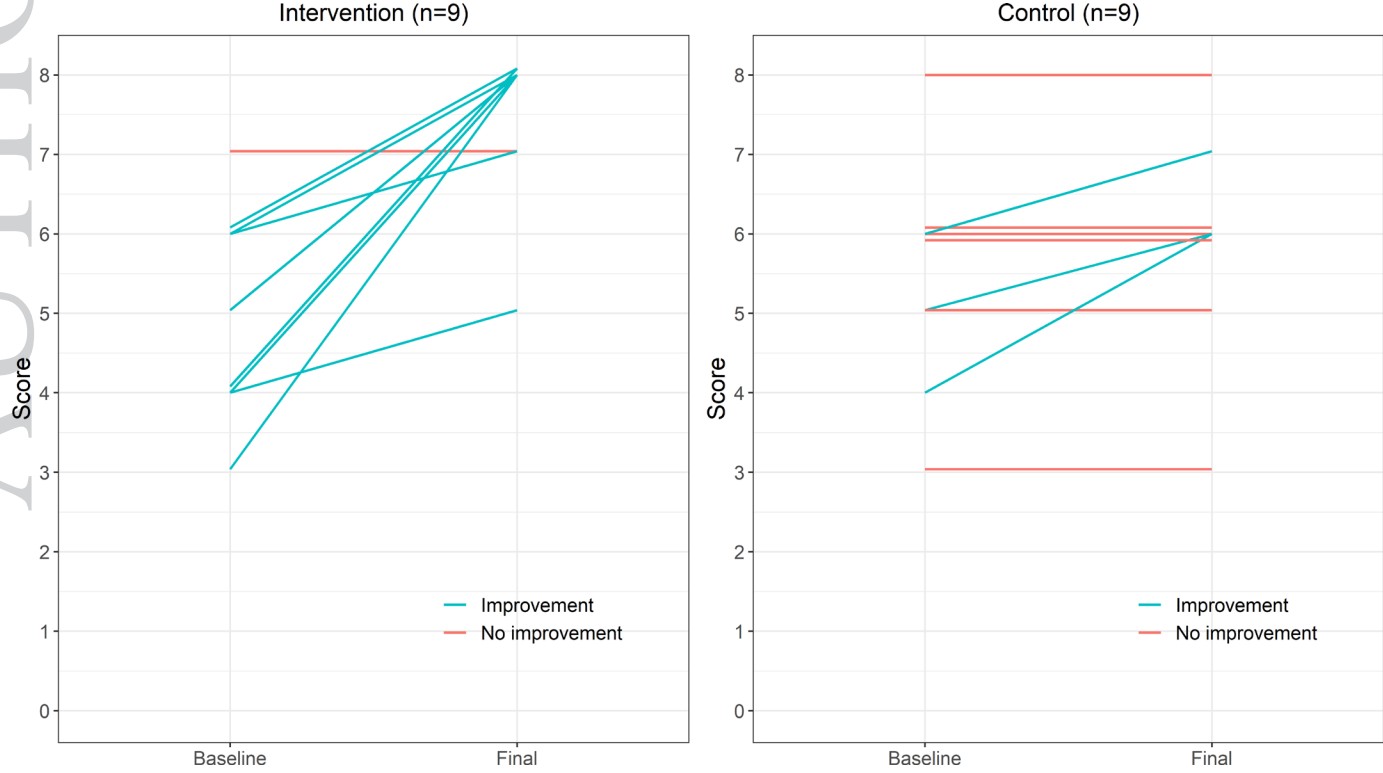

**Figure 3** Evolution of the scores for all manuscripts that passed the first round of peer review (n=18).

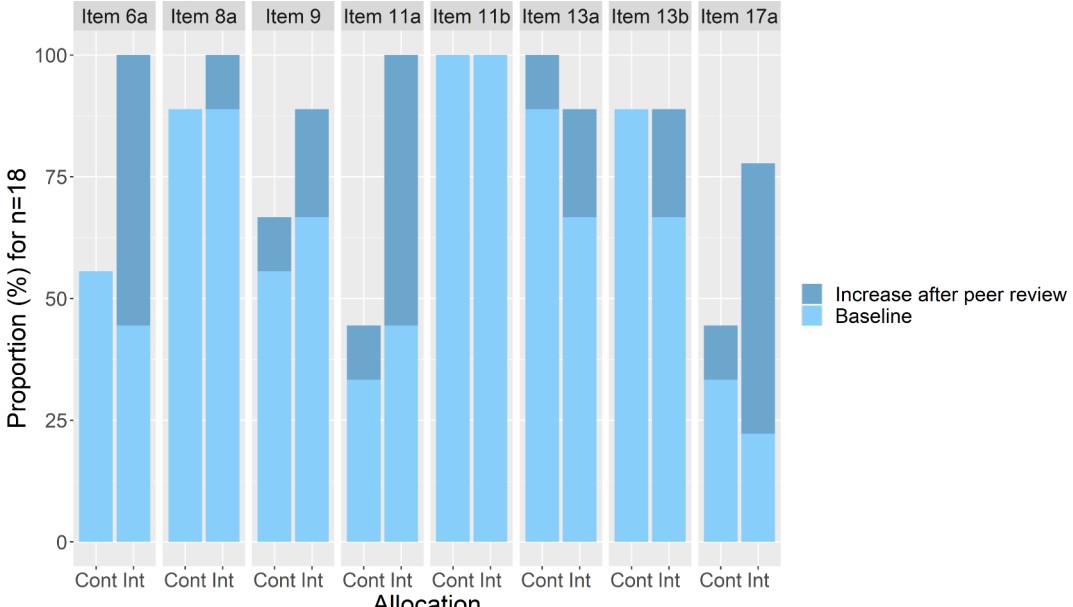

**Figure 4** Proportion of manuscripts (n=18) where each CONSORT item is adequately reported. CONSORT items: 6a: 'Completely defined pre-specified primary and secondary outcome measures, including how and when they were assessed'; 8a: 'Method used to generate the random allocation sequence'; 9: 'Mechanism used to implement the random allocation sequence (such as sequentially numbered containers), describing any steps taken to conceal the sequence until interventions were assigned'; 11a: 'If done, who was blinded after assignment to interventions (for example, participants, care providers, those assessing outcomes) and how'); 11b: 'If relevant, description of the similarity of interventions'; 13a: 'For each group, the numbers of participants who were randomly assigned, received intended treatment, and were analysed for the primary outcome'; 13b: 'For each group, losses and exclusions after randomisation, together with reasons'; 17a: 'For each primary and secondary outcome, results for each group, and the estimated effect size and its precision (such as 95% confidence interval)'). CONSORT, Consolidated Standards of Reporting Trials; Cont, control group; Int, intervention group.

### Secondary outcome

Figure 4 displays the proportions of manuscripts where each CONSORT item was adequately reported. We observed the main differences favouring the intervention group in items 6a (outcomes), 9 (allocation concealment mechanism), 11a (blinding) and 17a (outcomes and estimation).

### Feasibility of the intervention

The mean (SD) time taken to perform the intervention was 87 (42) min. Online supplementary file 4 displays a scatter plot that compares the amount of time spent to perform the intervention and the baseline score of the 12 manuscripts in the intervention group. There was no correlation between these two variables (r=0.08).

### Harms

We did not identify any unintended effects. There were no differences between the intervention and the control groups for the proportion of manuscripts that were rejected after the first round of peer review (3 of 12, 25%, for each group). Furthermore, all authors submitted the revised manuscripts within the period requested by the handling editor.

### DISCUSSION

We found that the introduction during the peer review process of an editorial intervention performed by a researcher with expertise in the content of CONSORT significantly improved the completeness of reporting of trials submitted to *BMJ Open* compared with standard peer review. Six of the nine manuscripts in the intervention group achieved the maximum score and another two improved. In contrast, the only manuscript in the control group with the maximum score at follow-up already had reached that score at baseline. We observed the main differences favouring the intervention group in items 6a (outcomes), 9 (allocation concealment mechanism), 11a (blinding) and 17a (outcomes and estimation). Moreover, providing authors with extra comments on reporting issues did not seem to discourage them from revising the manuscript as all authors returned the revised manuscripts within the standard 28 days requirement.

### Strengths and limitations

This study has several strengths: the randomised trial design; the fact that the intervention was performed in a real editorial context alongside peer review reports with no disruption to usual editorial procedures; and the fact that the outcome assessment process was blinded and in duplicate.

We also note some limitations that affect the generalisability of our results. Our intervention was focused only on CONSORT, which is one of the most well-established RGs. It could potentially be more difficult for authors to fully address reviewers' comments about other less

familiar RGs. We only included one journal and the same effect might not be observed in other journals. Nonetheless, we purposefully selected a very large general medical journal receiving international submissions across multiple specialties. We considered only eight core CONSORT items that are essential for evaluating the risk of bias of RCTs and not the whole checklist.

## Implications

Given the importance of improving the completeness of reporting of randomised trials and given the ineffectiveness of the strategies that biomedical journals are currently implementing,[13–16] it is time to take a step forward. Our study provides empirical evidence of the effectiveness of involving in the peer review process a researcher with expertise in CONSORT. In this study, the intervention was carried out by a PhD student and was implemented alongside peer review. However, this intervention could potentially be done by trained editorial staff, editors or external consultants. The demonstrated benefits of our intervention should encourage journal editors to find the best way to make this feasible.

We note that the complete case analysis and the best-case scenario of the sensitivity analysis point to a larger effect of the intervention than the main analysis. The worst-case scenario shows no effect. However, this scenario would assume (1) that the three rejected manuscripts in the intervention group would not improve from baseline; and (2) that all manuscripts in the control group would reach the maximum score. This scenario seems highly unlikely given that eight out of nine manuscripts that were not rejected in the intervention group improved from baseline and that only three controls improved and none of these reached the maximum score.

More than two decades ago, scientists started to discuss the importance of including statistical reviews as part of the publication process.[36] Nowadays, statistical reviews have become widespread among top medical journals. These are usually performed by a statistician and focus on the methodological and statistical aspects of the study. As methodological issues are often not fixable, statistical reviews are key to determining the fate of manuscripts and preventing unsound research getting published.[37] Completeness of reporting reviews should also become a key component in the publication system. As reporting issues are often improvable, these reviews should not generally aim to determine whether a manuscript should be published or not, but to improve their transparency. This would both help editors and peer reviewers make decisions on the manuscripts and improve the usability of published papers.

A few other RCTs have assessed different strategies for improving adherence to RGs. A recent RCT did not show that requesting authors to submit a checklist improves completeness of reporting and called for more stringent editorial policies.[16] The implementation of a writing aid tool for authors (CONSORT-based WEB tool) led to a moderate improvement in the completeness of

reporting,[11] whereas getting a statistician to perform an additional review against RGs showed a slightly positive but smaller than hypothesised effect.[10] Suggesting peer reviewers to check RGs[9] and implementing the web-based tool WebCONSORT at the manuscript revision stage showed no positive impact.[12] However, comparisons between the results of our study and these RCTs must be made with caution as they targeted different RGs and were carried out in different settings.

The time taken for us to perform the intervention (87 min on average, with great variation between manuscripts) is clearly a barrier to wider implementation. Future research could evaluate whether this intervention should be focused on the whole CONSORT checklist, which would make this strategy even more time-consuming, or only on a few core items (such as those we found to be poorly reported). Also, it would be interesting to assess whether similar benefits can be obtained for other widely used RGs, such as the Standard Protocol Items: Recommendations for Interventional Trials[38] or the Preferred Reporting Items for Systematic Reviews and Meta-Analyses.[39] Furthermore, this intervention could also be tested at other points in the editorial process, for example before the first decision is made on the manuscript or between the first decision and the invitation of external peer reviewers. For this study, we discarded both options for pragmatic reasons, as we did not want to alter the usual editorial process. While the first could be too resource-intensive for journals, the latter would imply the same effort and the manuscript would undergo more transparent and accurate peer review, which could make the task of peer reviewers and handling editors easier and more efficient. We strongly recommend that journals always carry out experiments in real editorial contexts, such as this study, before considering making any changes in their policies.

## CONCLUSIONS

This study provides evidence that involving a researcher with expertise in CONSORT in the process of evaluating RG checklists submitted by authors significantly improves the completeness of reporting of randomised trials. This is essential to reducing the research waste associated within adequate reporting of RCT methods and findings. Journal editors should consider revising their peer review processes to find ways to make this intervention workable, tailoring it to their preferences.

**Acknowledgements** We thank the MiRoR Project and the Marie Sklodowska-Curie Actions for their support. We thank *BMJ Open* for collaborating with this project, and also José Antonio González and Jordi Cortés (Universitat Politècnica de Catalunya) for collaborating in the process of developing the R codes used to perform the randomisation and outcome analysis.

**Collaborators** José Antonio González and Jordi Cortés (Universitat Politècnica de Catalunya).

**Contributors** DB: conceptualisation, methodology, software, formal analysis, investigation, writing (original draft preparation). SS, AA: conceptualisation,

methodology, resources, writing (review and editing). DM, IB: conceptualisation, methodology, writing (review and editing). JJK, EC: conceptualisation, methodology, outcome evaluation, writing (review and editing), supervision.

**Funding** This study is part of the ESR 14 research project from the Methods in Research on Research (MiRoR) project (http://miror-ejd.eu/), which has received funding from the European Union's Horizon 2020 research and innovation programme under the Marie Sklodowska-Curie grant agreement no 676 207. DM is supported through a University Research Chair (University of Ottawa). The sponsor and the funding source had no role in the design of this study and will not have any role during its execution, analyses, interpretation of the data or decision to submit results.

**Competing interests** AA is Editor in Chief of *BMJ Open*. AA was involved in the design of the study and writing the manuscript but not in data collection or data analysis. AA was not involved in the decision-making on this manuscript; the handling editor for the manuscript was instructed to raise any queries to the Deputy Editor, and AA was blinded to the editorial notes and discussion of the manuscript. The editorial team were instructed not to treat this manuscript any differently and that they should reject it if the reviewers felt it was not methodologically robust. SS is Senior Researcher at The BMJ. DM is Director of the Canadian EQUATOR Centre. IB is Deputy Director of the French EQUATOR Centre. DM, IB and EC are members of the CONSORT steering group.

**Patient consent for publication** Not required.

**Ethics approval** We obtained ethics approval from the Research Committee of the Governing Council of the Universitat Politècnica de Catalunya (UPC) (ref: EC 02, 13 September 2018). We did not seek consent from authors as this was part of a quality improvement programme at *BMJ Open*. However, all authors of the submitted manuscripts are routinely informed that BMJ has a research programme and that they can opt out if they wish.

**Provenance and peer review** Not commissioned; externally peer reviewed.

**Data availability statement** No data are available. The content of the intervention reports reflecting reporting inconsistencies will appear as part of the peer review history of the manuscripts included in the study. However, in order to protect confidentiality, we are not releasing any data set including individual manuscript data or outcome data identifying the performance of individual participants.

**Author note** This RCT is the third part of DB's PhD project, the first part of which was a scoping review to identify and classify interventions to improve adherence to RGs[8] and the second part was a survey to explore biomedical editors' opinion on various editorial interventions to improve adherence to RGs.[18]

**ORCID iDs**
David Blanco http://orcid.org/0000-0003-2961-9328
Sara Schroter http://orcid.org/0000-0002-8791-8564
Adrian Aldcroft http://orcid.org/0000-0003-0106-720X
David Moher http://orcid.org/0000-0003-2434-4206

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

AUTHOR PROOF

