## [Reviewer comments · BMJ Open]

ARTICLE DETAILS

TITLE (PROVISIONAL)	Effect of an editorial intervention to improve the completeness of reporting of randomised trials: a randomised controlled trial
AUTHORS	Blanco, David; Schroter, Sara; Aldcroft, Adrian; Moher, David; Boutron, Isabelle; Kirkham, Jamie J.; Cobo, Erik

VERSION 1 – REVIEW

REVIEWER	Nadia Elia University Hospitals of Geneva, Geneva, Switzerland
REVIEW RETURNED	31-Jan-2020

GENERAL COMMENTS	General comments: Thank you for giving me the opportunity to read this manuscript. I think the topic of completeness and transparency of reporting of RCTs is very important. However, I must admit that I am a bit disappointed by the very small sample size analysed in the present manuscript (24 RCTs (12/group) of which 6 were lost to follow-up!), and the low number of consort items considered (8 of the 36 items and subitems). I do realise that increasing the sample size represents significant amount of work. In an editorial published in 2011 (Elia and Tramèr, EJA 2011; 28(7):478-80), I had the opportunity to evaluate the adherence to ALL CONSORT items in 50 RCTs (without implementing a response to the authors I agree) but, this is feasible. Also, I think a larger sample would increase both the interest and the potential generalisability of these findings. I have a problem with RCTs reporting the results of 2 groups of 9 “participants”. Major comments: 1) Sample size: The power analysis is not very clearly described. From what I understand from the text: in a «pilot study» based on 12 RCTs, the distribution of the RCTs according to the number of the 8 items correctly fulfilled was 0 RCT fulfilling 0 to 4 items, 2 RCTs correctly reporting 5 items, 4 correctly reporting 6 items, 4 correctly reporting 7 items and 2 that did so for 8 items (13%, 33%, 33% and 17%). The aim of the intervention is to bring this distribution to (0,0,0,0,0,0,50%, 50%). Then I see a mention of a 90% power to detect this difference, no mention of alpha level, no mention of one vs two-sided test. I am not a statistician, but I can think of different simple ways to compute a sample size requirement: a) to allow to see a statistically significant difference between 33% and 50%, with 90% power... this requires 350 RCTs, not 24.
---

b) Or, to compare means scores by groups (which is actually what is presented in the result section) : from mean 6.5 to 7.5, this would require 46 for a two-sided test, 36 for a one-sided test, 28 for a one sided test with 80% power... we're getting close, but not yet to 24.

c) Or consider the difference between the initial and final score in each group...

It remains unclear, for the average reader, what has been done. I saw the R code in the Appendix, it seems to be a power calculation based on the fact that 12 RCTs per group were already defined, but not a pre-hoc sample size estimation...? Anyways, I'm not sure this is transparent enough to be understood by the average reader. This should be clarified. Especially since this is the basis to justify a very small sample size.

2) Missing data:

When reading the Abstract only, it is not clear what is meant by : "We imputed missing data for the main analysis (n=24)"; You counted the items in the initial MS, then you counted the items in the revision, how can data be missing?

After reading the method section, I understand that the authors have considered as "missing data" when:

- o Editors rejected the a MS after peer-review, and therefore the authors were not able to correct/or not, the consort items.

- o Authors did not return the revised MS with the period requested I don't think these are "missing data". These are "lost to follow-up".

And should have been accounted for in the estimation of the sample size required. You don't know what would have happened if the authors had sent their MS back. Imputing 25% of the data regarding the outcome, in such a small sample seems a bit... hazardous. Why not present a "best case" (all authors have done all the requested changes) and "worst case" scenario (none of the authors have done the requested changes) ? Also, I don't understand the basis underlying how the outcome was imputed.

Minor comments:

The protocol for this study was registered which is good.

Abstract :

What do you mean by a « consort expert » ? Is this just someone trained to identify the presence or absence of CONSORT items in a scientific text ? How do you become an "expert" ?

Methods:

/Randomisation and blinding

There is at least one word missing in the first sentence «Every we detected a manuscript»

Statistical methods

Not sure why you "adjust for baseline score". Were the baseline scores so different between the groups? Why not use differences in scores per manuscript?

Results

I'm surprised that in a 5 months period, there were only 62 RCTs submitted to BMJOpen that underwent peer-review (" a very large general journal receiving international submissions across multiple specialities")? How many were rejected before peer-review ? A flow chart describing all MS submitted to BMJOpen during that

	period, the number excluded, with reasons, and the final number that fulfilled inclusion criteria etc... would be nice. Not sure why you excluded RCTs submitted "without CONSORT checklist". I suppose a "CONSORT expert" should be able to check the adherence to CONSORT recommendation, without a checklist. These checklist are rarely useful and do not reflect the information provided. I miss a baseline description of the RCT analysed (CONSORT pt 15): what were the topics covered by these RCTs? The origin of the authors? Multi or single -center trials ? Were they sponsored? By whom? What was the trial size? important for generalisability of the findings. Figure 3 should highlight the n=9 in each group. Figure 4: unclear which n is reported (24? 18?) Table 1: the numbers reported do not match those in the text (control 5.68 (1.79) in table, 5.68 (1.43) in the text and abstract) Authors report on " correlation between the amount of time taken and baseline score of the manuscript", I would be curious to see a scatter plot. Discussion One major limitation that should be discussed is the very small sample size. I am not sure that the impact (improvement of 1.4 / 8) of the intervention can be described as "major" (page 14 line 60). Especially when one considers that it takes, on average, 1h30 of an "expert's time" to achieve this. Ethics: in teh abstract, authors report that "authors of MS were unaware that they were part of an RCT". However on page 16, they report that "all authors of the submittes manuscript were informed that BMJ has research programme and that they could opt out if the wished"... although I can imagine what is described here, it seems a bit... weird. References This is an anecdotic point but, of the 40 references cited, 5 are internet links, and of the 35 remaining references, at least 25 are references authored by at least one co-author of the present manuscript. I do not know if BMJOpen has any specific rules regarding auto-citation.
--	---

REVIEWER	Gui-shuang Ying University of Pennsylvania Perelman School of Medicine, Ophthalmology
REVIEW RETURNED	20-Feb-2020

GENERAL COMMENTS	This paper reported the results from a small randomized clinical trial on the effect of editorial intervention to improve the completeness of report of clinical trial. The study is well conducted and the results are interesting. A few comments are made to improve the manuscript. 1. The sample size calculation for the trial came up with 12 manuscripts in each arm. There are 25% manuscript had missing data (due to rejection of paper). Was this attrition rate due to rejection of paper not considered in the initial sample size
--

	calculation? Also, the it was indicated 90% power in the manuscript, but was indicated as 85% power in the sample size section of supplement document. This inconsistency should be corrected. 2. In page 12, lines32-36 described the approach for imputing missing data. It is not clear why the missing data was imputed as 1-b (where b is the baseline score of manuscript). Is the b the overall total score or the score of each item? This needs to be made clear. If it is for the total score, the baseline score ranges from 0 to 8, using 1-b does not make sense. 3. Table 1: It will be informative to add their baseline score, so that readers can get sense on how much improvement after intervention. It will also be informative to report the secondary outcomes in this table.
--	---

REVIEWER	Sabrina Tulka Institute for Medical Biometry and Epidemiology, Faculty of Health, Witten/Herdecke University Germany
REVIEW RETURNED	26-Feb-2020

GENERAL COMMENTS	It is obvious that the simple existence of the CONSORT checklist is not sufficient to ensure the publication of complete and transparent study descriptions. It is absolutely right that this problem, in particular in the peer review process and thus, of course, before a manuscript is published, must be resolved in order to guarantee good and complete study descriptions. Thus, the author's approach to directly address deficiencies in the consideration or implementation of CONSORT's in the review process is a good idea. However, some questions or comments came up while reading the manuscript: Why did the authors not use the whole CONSORT checklist but only eight core items? Why were items 6a, 8a, 9, 11a, 11b, 13a, 13b, 17a chosen in particular? Are there reasons to include only and explicitly these items? Reasons for the choice are missing in the manuscript. Some points of the CONSORT list may be considered less important, but I personally miss the sample size calculation as a core item. Here we detected deficiencies in terms of content and methodology, while this gives the ethical justification for the number of patients included in a RCT, it is rarely reported and in most cases, if at all, incorrect or incomplete. (Tulka et al. (2019) Validity of sample sizes in publications of randomised controlled trials on the treatment of age-related macular degeneration: cross-sectional evaluation BMJ Open. 2019 Oct 10; 9 (10): e030312. doi: 10.1136/bmjopen-2019-030312). Therefore I would recommend the authors to include sample size calculation in their assessment and review (intervention) process. I demand at least a discussion why sample size calculation is missing as a key item. Are you planning a larger study (with all CONSORT items)? I'm missing that information in the discussion of your paper. I guess that you plan to include all items when your idea is implemented in real reviews?
---

	Why did you only include manuscripts that provided a CONSORT checklist and therefore included CONSORT in their manuscript? Following CONSORT would have to be demanded of everyone and especially of authors who did not submit the CONSORT checklist. Individual differences for each publications in both groups would be interesting in order to be able to see the effect of the intervention a better (directly in numbers and not only in Figure 3) and in particular also the difference between the mean of these values in the intervention and the control group would be interesting and enable the reader to compare the groups easier. It would be interesting to know the different areas of indication and, for example, whether it is about drug studies, medical device studies, etc. in order to be able to assess how similar the two groups were on the one hand and to see whether there are specialist areas, that do particularly well or particularly badly in working with CONSORT. Are there other papers on this subject or on other strategies on how to induce authors to better comply with the CONSORT statement? If so, a comparison to other strategies would be desirable in the discussion section of the manuscript.
--	--

REVIEWER	Akihiro Nishi UCLA Epidemiology, USA
REVIEW RETURNED	02-Mar-2020

GENERAL COMMENTS	Conceptually, this is a simple and great paper. There are two questions. The authors collaborated with BMJ Open for this project. In addition, the Editor-in-Chief Adrian Aldcroft (AA) is one of the authors of the present manuscript. Nonetheless, the possibility that such a potential conflict of interest distorts the process of study design, data analysis, and manuscript writing has not been discussed. "AA is Editor in Chief of BMJ Open" in the Declaration of interests is not enough. At least, the manuscript should be evaluated by a different editorial team (i.e. a different journal). The authors aims to use BMJ Open for data collection, asked EIC to perform a RCT at BMJ Open, included EIC in the author team, and aim to publish a paper in BMJ Open - this reviewer feels this is simply too much. For this perspective, BMJ is a better place than BMJ Open for publication. (This may not be the question for the authors though. Rather, this can be a question for the Editorial policy of BMJ Open). The second question is informed consent: did the authors of the original RCT papers inform of their participation in this RCT before the random assignment? And, this is not a double-blinded RCT.
---

VERSION 1 – AUTHOR RESPONSE

Reviewer Name: Nadia Elia

Institution and Country: University Hospitals of Geneva, Geneva, Switzerland

Please state any competing interests or state 'None declared': Indirect conflicts of interest: I have been working since the last 10 years as "methods editor" of a speciality journal with the aim of improving the quality and reporting of published articles. No financial conflict of interest.

We would like to thank you for your detailed review and your many proposals to improve the quality of the manuscript. Please find below our point-by-point response to your comments.

Major comments:

1) Sample size:

The power analysis is not very clearly described. From what I understand from the text: in a «pilot study» based on 12 RCTs, the distribution of the RCTs according to the number of the 8 items correctly fulfilled was 0 RCT fulfilling 0 to 4 items, 2 RCTs correctly reporting 5 items, 4 correctly reporting 6 items, 4 correctly reporting 7 items and 2 that did so for 8 items (13%, 33%, 33% and 17%). The aim of the intervention is to bring this distribution to (0,0,0,0,0,0,50%, 50%). Then I see a mention of a 90% power to detect this difference, no mention of alpha level, no mention of one vs two-sided test.

I am not a statistician, but I can think of different simple ways to compute a sample size requirement: a) to allow to see a statistically significant difference between 33% and 50%, with 90% power... this requires 350 RCTs, not 24.

b) Or, to compare means scores by groups (which is actually what is presented in the result section) : from mean 6.5 to 7.5, this would require 46 for a two-sided test, 36 for a one-sided test, 28 for a one-sided test with 80% power... we're getting close, but not yet to 24.

c) Or consider the difference between the initial and final score in each group...

It remains unclear, for the average reader, what has been done. I saw the R code in the Appendix, it seems to be a power calculation based on the fact that 12 RCTs per group were already defined, but not a pre-hoc sample size estimation...? Anyways, I'm not sure this is transparent enough to be understood by the average reader. This should be clarified. Especially since this is the basis to justify a very small sample size.

We have re-written the "Power analysis" section to describe the procedure used to calculate the study power more clearly.

2) Missing data:

When reading the Abstract only, it is not clear what is meant by: "We imputed missing data for the main analysis (n=24)"; You counted the items in the initial MS, then you counted the items in the revision, how can data be missing?

After reading the method section, I understand that the authors have considered as "missing data" when:

- o Editors rejected the a MS after peer-review, and therefore the authors were not able to correct/or not, the consort items.

- o Authors did not return the revised MS with the period requested

I don't think these are "missing data". These are "lost to follow-up". And should have been accounted for in the estimation of the sample size required. You don't know what would have happened if the authors had sent their MS back. Imputing 25% of the data regarding the outcome, in such a small sample seems a bit... hazardous. Why not present a "best case" (all authors have done all the requested changes) and "worst case" scenario (none of the authors have done the requested changes)? Also, I don't understand the basis underlying how the outcome was imputed.

Regarding readability, we have made the Abstract clearer regarding what we mean by missing data.

We have also used "lost to follow-up" in the text to be more precise about the kind of missing data we are referring to.

Regarding methods, please note that our main analysis of the primary outcome is "intention-to-treat" (following CONSORT wording and recommendations), as specified in the protocol. This analysis takes into consideration all manuscripts, regardless of the editorial decision (whether they had been

rejected or not). For this reason, the attrition rate does not need to be taken into account for the sample size calculation.

We have also fixed an error regarding the values we used to impute the scores of lost to follow-up manuscripts. It is $8-b$, where b is the baseline score, instead of $1-b$.

As you suggested, we have also added in the text and in Table 1 (now Table 2) the best- and worst-case scenario analysis to assess the sensitivity of results.

Minor comments:

Abstract: What do you mean by a « consort expert » ? Is this just someone trained to identify the presence or absence of CONSORT items in a scientific text? How do you become an “expert”? We have clarified that point in the Abstract. Given the word count restriction in the Abstract, we expand on it in the “Intervention” section.

Methods: /Randomisation and blinding

There is at least one word missing in the first sentence «Every we detected a manuscript»

We have reworded that sentence to make it clearer.

Statistical methods: Not sure why you “adjust for baseline score”. Were the baseline scores so different between the groups? Why not use differences in scores per manuscript?

We have now included the baseline scores for the two groups in Table 1 (now Table 2). We adjusted for baseline scores because we tried to avoid that a difference in the baseline scores between the two groups could make the intervention seem to have a larger or smaller effect than it actually had. We have pointed that out in the “Deviations from the protocol” section.

Results: I’m surprised that in a 5 months period, there were only 62 RCTs submitted to BMJOpen that underwent peer-review (“a very large general journal receiving international submissions across multiple specialities”)? How many were rejected before peer-review? A flow chart describing all MS submitted to BMJOpen during that period, the number excluded, with reasons, and the final number that fulfilled inclusion criteria etc... would be nice.

As explained in the first paragraph of the Results section, 62 RCTs were submitted to BMJ Open but most were rejected before undergoing peer review. Figure 2, the CONSORT flow diagram for our study, already displays all the data you have suggested. Despite the fact that BMJ Open is a very large medical journal, it receives only a small number of reports of RCTs. This pattern is seen throughout the year and is not a result of only studying manuscripts between X and Y dates.

Not sure why you excluded RCTs submitted “without CONSORT checklist”. I suppose a “CONSORT expert” should be able to check the adherence to CONSORT recommendation, without a checklist. These checklist are rarely useful and do not reflect the information provided.

BMJ-Open “Instructions to authors” mention that, when applicable, manuscripts need to include a RG checklist. We added this eligibility criterion just in case the editorial staff did not realise that the checklist was missing. Furthermore, our intervention aimed to be an attempt to improve completeness by using the submitted checklists. For this reason, if a checklist was not provided we could not perform the intervention. We had already mentioned that point in the “Deviations from the protocol” section. We have improved the description of the intervention (“Intervention” section) to make clear that we needed the checklist to perform the intervention.

I miss a baseline description of the RCT analysed (CONSORT pt 15): what were the topics covered by these RCTs? The origin of the authors? Multi or single -center trials ? Were they sponsored? By whom? What was the trial size? important for generalisability of the findings.

We have included a new Table 1 specifying the baseline characteristics of the included RCTs.

Figure 3 should highlight the $n=9$ in each group.

We have included that information in Figure 3.

Figure 4: unclear which n is reported (24? 18?)

We have included in Figure 4 that n is also 18.

Table 1: the numbers reported do not match those in the text (control 5.68 (1.79) in table, 5.68 (1.43) in the text and abstract)

The SD reported in Table 1 (now Table 2) was wrong and we have corrected it.

Authors report on "correlation between the amount of time taken and baseline score of the manuscript", I would be curious to see a scatter plot.

We have added a scatter plot reporting this information (Supplementary file 4).

Discussion

One major limitation that should be discussed is the very small sample size.

I am not sure that the impact (improvement of 1.4 / 8) of the intervention can be described as "major" (page 14 line 60). Especially when one considers that it takes, on average, 1h30 of an "expert's time" to achieve this.

Sample size, n , is linked to the estimated effect size in our pilot calculations, δ . As we estimated a major effect size, a reduced sample size of $n=24$ papers was enough to guarantee a 90% power ($\alpha 0.05$, two-sided). Furthermore, from a pragmatic side we believed that this was a reasonable number to achieve in around half a year and that it implied a reasonable workload for outcome assessors. Due to the small number of RCT submissions and even smaller number of RCTs that undergo peer review, reaching a sample size of 100 RCT could take around 2 years.

We also believe that the effect of the intervention could be described as large, according to the categorisation by Cohen, 1988. Cohen's D for our study is 0.91 ($= 1.43/1.57$). Moreover, in Figure 3 you can see that six of the nine manuscripts in the intervention group achieved the maximum score and other two improved. In contrast, the only manuscript in the control group that reached the maximum score already had that score at baseline.

Ethics:

in the abstract, authors report that "authors of MS were unaware that they were part of an RCT".

However on page 16, they report that "all authors of the submitted manuscript were informed that BMJ has research programme and that they could opt out if they wished"... although I can imagine what is described here, it seems a bit... weird.

We have clarified this point in "Randomisation and blinding" section. As usual, authors were informed that BMJ Publishing Group has a quality improvement programme but they were not explicitly told that their manuscripts were part of an RCT.

References

This is an anecdotal point but, of the 40 references cited, 5 are internet links, and of the 35 remaining references, at least 25 are references authored by at least one co-author of the present manuscript. I do not know if BMJ Open has any specific rules regarding auto-citation.

There are not many studies that have evaluated interventions to improve adherence to reporting guidelines nor many researchers working on this field. Most co-authors of this study are part of the Methods in Research on Research program (MiRoR), one of the most ambitious initiatives in Europe in the field of meta-research. We have included all key references in this under researched area. In addition, BMJ Open does not specify any rules regarding self-citation.

Reviewer: 2

Reviewer Name: Gui-shuang Ying

Institution and Country: University of Pennsylvania Perelman School of Medicine, Philadelphia, PA 19104, USA

We thank you for taking the time to read and comment on our paper. Please find below our point-by-point response to your comments.

1. The sample size calculation for the trial came up with 12 manuscripts in each arm. There are 25% manuscript had missing data (due to rejection of paper). Was this attrition rate due to rejection of paper not considered in the initial sample size calculation?

Please note that our main analysis of the primary outcome is "intention-to-treat" (following CONSORT wording and recommendations), as specified in the protocol. This analysis takes into consideration all manuscripts, regardless of the editorial decision (whether they had been rejected or not). For this reason, the attrition rate does not need to be taken into account for the sample size calculation.

Also, the it was indicated 90% power in the manuscript, but was indicated as 85% power in the sample size section of supplement document. This inconsistency should be corrected. We have clarified this point in the “Deviations from the protocol” section. The initial study power specified in the protocol was 85%. However, after deciding to use bootstrapping technique for the primary outcome analysis (in order not to rely on t-test required assumptions for a small sample size), the power increased to 90%, which is the number shown in the “Power analysis” section. We have also re-written the “Power analysis” section to make the procedure used to calculate the study power clearer.

2. In page 12, lines32-36 described the approach for imputing missing data. It is not clear why the missing data was imputed as 1-b (where b is the baseline score of manuscript). Is the b the overall total score or the score of each item? This needs to be made clear. If it is for the total score, the baseline score ranges from 0 to 8, using 1-b does not make sense.

We have corrected that mistake and made it clear that missing data were imputed as 8-b, where b is the baseline score of the manuscript. Furthermore, we have included a sensitivity analysis showing the best- and worst-case scenarios.

3. Table 1: It will be informative to add their baseline score, so that readers can get sense on how much improvement after intervention. It will also be informative to report the secondary outcomes in this table.

We have added the baseline scores for the two groups to Table 1 (now Table 2). For the secondary outcome of the study (Proportions of manuscripts where each CONSORT item was adequately reported), we believe that Figure 4 is clearer and more illustrative than including the raw data in Table 1 (now Table 2). Of course, we could add this data as secondary online material if you still think it is required.

Reviewer: 3

Reviewer Name: Sabrina Tulka

Institution and Country: Institute for Medical Biometry and Epidemiology, Faculty of Health, Witten/Herdecke University, Germany

We would like to thank you for your detailed review. Please find below our point-by-point response to your comments.

Why did the authors not use the whole CONSORT checklist but only eight core items? Why were items 6a, 8a, 9, 11a, 11b, 13a, 13b, 17a chosen in particular? Are there reasons to include only and explicitly these items? Reasons for the choice are missing in the manuscript.

These 8 items are key for researchers doing systematic reviews to help them evaluate the risk of bias of RCTs. We did mention why we only included these items in the “Interventions” section, just before the description of each item’s content. However, we have now rephrased that sentence to make this point clearer and also mentioned it in the limitations section.

Some points of the CONSORT list may be considered less important, but I personally miss the sample size calculation as a core item. Here we detected deficiencies in terms of content and methodology, while this gives the ethical justification for the number of patients included in a RCT, it is rarely reported and in most cases, if at all, incorrect or incomplete. (Tulka et al. (2019) Validity of sample sizes in publications of randomised controlled trials on the treatment of age-related macular degeneration: cross-sectional evaluation *BMJ Open*. 2019 Oct 10; 9 (10): e030312. doi: 10.1136/bmjopen-2019-030312). Therefore I would recommend the authors to include sample size calculation in their assessment and review (intervention) process. I demand at least a discussion why sample size calculation is missing as a key item.

As mentioned above, we did not include the item related to sample size, despite the fact that it could be of interest, because we aimed to focus on the key items used to evaluate the risk of bias of RCTs. We have clarified this point clearer in the “Intervention” and “Discussion” sections.

Are you planning a larger study (with all CONSORT items)? I'm missing that information in the discussion of your paper. I guess that you plan to include all items when your idea is implemented in real reviews?

We have re-written most of the last paragraph of the Discussion section in order to be more precise on the implications of our study and what questions we would like future research to address.

Why did you only include manuscripts that provided a CONSORT checklist and therefore included CONSORT in their manuscript? Following CONSORT would have to be demanded of everyone and especially of authors who did not submit the CONSORT checklist.

BMJ-Open "Instructions to authors" mention that, when applicable, manuscripts need to include a RG checklist. We added this eligibility criterion just in case the editorial staff did not realise that the checklist was missing. Our intervention aimed to be an attempt to improve completeness by using the submitted checklists. For this reason, if a checklist was not provided we did not perform the intervention.

Individual differences for each publications in both groups would be interesting in order to be able to see the effect of the intervention a better (directly in numbers and not only in Figure 3) and in particular also the difference between the mean of these values in the intervention and the control group would be interesting and enable the reader to compare the groups easier.

We have presented the mean difference in scores between the two groups in the "Primary Outcome" section. For the individual scores of each paper, we believe that displaying them in Figure 3 is clearer and more illustrative than enumerating them.

It would be interesting to know the different areas of indication and, for example, whether it is about drug studies, medical device studies, etc. in order to be able to assess how similar the two groups were on the one hand and to see whether there are specialist areas, that do particularly well or particularly badly in working with CONSORT.

We have included a new Table 1 specifying the baseline characteristics of the included RCTs.

We did not plan any subgroup analysis (see protocol) and so none are reported. We have indicated it in the "Statistical methods" section.

Are there other papers on this subject or on other strategies on how to induce authors to better comply with the CONSORT statement? If so, a comparison to other strategies would be desirable in the discussion section of the manuscript.

We have added a paragraph on the Discussion section comparing these results to other strategies.

Reviewer: 4

Reviewer Name: Akihiro Nishi

Institution and Country: UCLA Epidemiology, USA

We thank you for taking the time to read and comment on our paper. Please find below our point-by-point response to your comments.

The authors collaborated with BMJ Open for this project. In addition, the Editor-in-Chief Adrian Aldcroft (AA) is one of the authors of the present manuscript. Nonetheless, the possibility that such a potential conflict of interest distorts the process of study design, data analysis, and manuscript writing has not been discussed. "AA is Editor in Chief of BMJ Open" in the Declaration of interests is not enough. At least, the manuscript should be evaluated by a different editorial team (i.e. a different journal). The authors aims to use BMJ Open for data collection, asked EIC to perform a RCT at BMJ Open, included EIC in the author team, and aim to publish a paper in BMJ Open - this reviewer feels this is simply too much. For this perspective, BMJ is a better place than BMJ Open for publication. (This may not be the question for the authors though. Rather, this can be a question for the Editorial policy of BMJ Open).

We have made it clear in the Abstract that this study is something BMJ Open evaluated as part of the journal's quality improvement programme. In the conflicts of interest section, we have included the

following information: “AA was involved in the design of the study and writing the manuscript but not in data collection or data analysis. AA was not involved in the decision-making on this manuscript; the handling editor for the manuscript was instructed to raise any queries to the Deputy Editor and AA was blinded to the editorial notes and discussion of the manuscript. The editorial team were instructed not to treat this manuscript any differently and that they should reject it if the reviewers felt it was not methodologically robust.”

The second question is informed consent: did the authors of the original RCT papers inform of their participation in this RCT before the random assignment? And, this is not a double-blinded RCT. In order to make this point clearer, we have added some information in the “Randomisation and blinding” section. All submitting authors and invited reviewers are routinely informed that BMJ Publishing Group has a quality improvement programme and that their manuscript/review may be entered into a study but the authors in this study were not explicitly told that their manuscripts were part of an RCT.

VERSION 2 – REVIEW

REVIEWER	Gui-shuang Ying University of Pennsylvania
REVIEW RETURNED	07-Apr-2020

GENERAL COMMENTS	The authors address all the previous comments satisfactorily.
---

REVIEWER	Sabrina Tulka Institute for Medical Biometry and Epidemiology, University Witten Herdecke Faculty of Health, Witten, Germany
REVIEW RETURNED	06-Apr-2020

GENERAL COMMENTS	I'm still missing the discussion of some important limitations of the presented work. The authors should mention, discuss and explain that (and why) they only analysed so called „key items“. In my opinion some aspects of the CONSORT Statement would be more important to check than some of the „key items“. The authors should at least tell us how and why they have chosen only a small part of CONSORT to improve. This aspect must be included in the limitations section of the presented work.
--